# Synergistic Anticancer Activity of *N*-Hydroxy-7-(2-Naphthylthio) Heptanomide, Sorafenib, and Radiation Therapy in Patient-Derived Anaplastic Thyroid Cancer Models

**DOI:** 10.3390/ijms22020536

**Published:** 2021-01-07

**Authors:** Hyeok Jun Yun, Hee Jun Kim, Jungmin Kim, Sang Yong Kim, Hang-Seok Chang, Cheong Soo Park, Ho-Jin Chang, Ki Cheong Park

**Affiliations:** 1Department of Surgery, Gangnam Severance Hospital, Yonsei University College of Medicine, 211 Eonjuro, Gangnam-gu, Seoul 135-720, Korea; GSYHJ@yuhs.ac (H.J.Y.); KHJ9792DR@yuhs.ac (H.J.K.); SURGHSC@yuhs.ac (H.-S.C.); CSPARK1@yuhs.ac (C.S.P.); 2Department of Surgery, Yonsei University College of Medicine, 50-1, Yonsei-ro, Seodaemun-gu, Seoul 120-752, Korea; JM_KIM@yuhs.ac (J.K.); 0101YONG@yuhs.ac (S.Y.K.)

**Keywords:** patient-derived anaplastic thyroid cancer, *N*-hydroxy-7-(2-naphthylthio) heptanomide, sorafenib, radiation therapy, apoptosis

## Abstract

Anaplastic thyroid cancer (ATC) is an undifferentiated and advanced form of thyroid cancer, accompanied with a high ratio of epigenetic adjustment, which occurs more than genetic mutations. In this study, we aimed to evaluate the synergistic anticancer effect (in vitro and in vivo) of the new combination of *N*-hydroxy-7-(2-naphthylthio) heptanomide (HNHA) and sorafenib with radiation therapy in pre-clinical models of ATC. The ATC cell lines, YUMC-A1 and YUMC-A2, were isolated from the current patients who were treated with HNHA and sorafenib, either as monotherapy or combination therapy. Synergistic anticancer effect of the combination therapy on the intracellular signaling pathways and cell cycle was assessed via flow cytometry and immunoblot analysis. To examine tumor shrinkage activity in vivo, an ATC cell line-derived mouse xenograft model was used. Results showed that the combination therapy of HNHA and sorafenib with radiation promoted tumor suppression via caspase cleavage and cell cycle arrest in patient-derived ATC. In addition, the combination therapy of HNHA and sorafenib with radiation was more effective against ATC than therapy with HNHA or sorafenib with radiation. Thus, the combination of HNHA and sorafenib with radiation may be used as a novel curative approach for the treatment of ATC.

## 1. Introduction

The thyroid gland secretes thyroid hormones that play decisive roles in managing normal metabolism [1,2,3]. Thyroid cancer is classified into four main types, as follows: anaplastic, papillary, follicular, and medullary [4,5]. Thyroid cancer is also categorized as either well-differentiated or poorly differentiated/anaplastic, based on the characteristics of cell differentiation and maintenance of the follicular cell phenotype. Differentiated thyroid cancer is the most common type of thyroid cancer, accounting for more than 80%–90% of all thyroid cancers and comprises papillary and follicular histological subtypes [6,7]. In contrast, the incidence of poorly differentiated or anaplastic thyroid cancer (ATC) is just 2–3% of clinically recognized thyroid cancer cases. However, ATC has a poor prognosis and demonstrates chemo-therapeutic resistance and aggressiveness [8], and the median overall survival (OS) is limited to few months [9,10,11]. Most of the patients with ATC are elderly and require effective treatment to control the swiftly growing tumor mass. Moreover, nearly 50% of patients require pain medication due to distant metastases [12,13,14]. In most of the patients, complete surgical resection is not feasible. As with this study, studies based on patient-derived ATC cell lines indicate promise that new clinical approaches to the management of ATC will be found in the future. Recent studies have identified various molecules and mechanisms that are precisely associated with the poor clinical results in ATC [15,16]. Among these mechanisms, the synergistic anticancer effect of the histone deacetylase (HDAC) inhibitor or sorafenib and radiation-induced suppression of advanced cancer has been considered one of the probable reasons for poor clinical results [17,18,19]. The positive efficacy results obtained with HDAC inhibitors as anticancer agents resulted in the approval of HDAC inhibitors by the US Food and Drug Administration (USFDA) for the treatment of some cancer subtypes [20,21]. Several HDAC inhibitors are presently used clinically as therapeutic drugs, either agents alone or combination with other anticancer agents [19,22,23]. *N*-hydroxy-7-(2-naphthylthio) heptanomide (HNHA) is an unusual HDAC inhibitor that has been proved to have significantly higher anticancer activity than trichostatin A (TSA) and suberoylanilide hydroxamic acid (SAHA) [24,25,26]. In addition, the USFDA recently authorized the use of sorafenib for ATC treatment [27]. Sorafenib is a multi-kinase inhibitor that interrupts diverse signaling pathways, including platelet-derived growth factor receptors (PDGFRs), vascular endothelial growth factor receptor (VEGFR), and Raf kinases [28]. Additionally, sorafenib has also been approved for the treatment of advanced renal cell carcinoma (RCC) and other human cancers [29,30,31,32].

In this study, we aimed to investigate a new clinical approach, the efficacy of the combination therapy of HNHA and sorafenib with radiation in ATC cell lines and in vivo mouse models of ATC.

## 2. Results

### 2.1. Patient Disease Characteristics

Patient demographics and disease characteristics are presented in Figure 1A. The mean age of the patients was 63.6 years, and 60.9% of patients were females. Stage IVB ATC and stage IVC ATC were diagnosed in 26.1 and 73.9% of the patients, respectively, and 45.7% of patients underwent surgery, while 30.5% underwent complete resection. All patients received chemotherapy with paclitaxel, and three patients previously received adriamycin at another medical institution, and 93.5% of patients received radiation therapy. The median OS for the 46 patients with ATC was 228 days (rage 58–1418) (Figure 1A). Survival rate was 63.2% at 6 months and 23.9% at 1 year (Figure 1B). For this reason, we researched more highly in differentiated than undifferentiated, ATC.

### 2.2. Characteristics of Patient-Derived Thyroid Cancer Cell Lines

Various subtypes of patient-derived differentiated and undifferentiated thyroid cancer cell lines were acquired from the patient specimens (Figure 2A). YUMC-A1 (first isolated patient-derived ATC) and YUMC-A2 (second isolated patient-derived ATC) were obtained from patients with ATC treated at the Severance Hospital, Yonsei University College of Medicine, Seoul, Korea. Therapy failure, cancer recurrence, and metastasis were reported in these patients. The results of the next-generation RNA sequencing to identify a series of differentially expressed genes (DEGs) showed that YUMC-A1 and YUMC-A2, patient-derived ATC (undifferentiated) cells, were associated with significantly increased levels of cancer stemness (*CD44^high^/CD24^low^, CDH2, KRAS CDC42, RAC1, MCPH1, SRGAP2, SRGAP2C, TWIST1, TWIST2, PROM1, ICAM1*) and metastatic markers (*BRACA1/2, KRT5, CD44, WNT4/16, ALDH1A1, CD29, SALL4*) when compared with patient-derived differentiated thyroid cancer cells, YUMC-F1 (first isolated patient-derived follicular thyroid cancer), YUMC-M1 (first isolated patient-derived medullary thyroid cancer), and YUMC-P1 (first isolated patient-derived papillary thyroid cancer), as shown in Figure 2B,C. In ATC, the most significantly upregulated genes were *BRACA2, CD44, CDC42*, and *RAC1*. The top four upregulated and downregulated DEGs are shown in Figure 2C. Taken together, we concluded that study of the ATC could be of great value to therapeutic trials in the management of patients with stemness and metastatic cancer, including drug-resistant properties.

### 2.3. Combination of HNHA and Sorafenib with Radiation Was More Effective Than Either HNHA or Sorafenib with Radiation

To evaluate the synergistic anticancer effects of the combination of HNHA and sorafenib on patient-derived ATC cells, we tested the proliferation of YUMC-A1 and YUMC-A2 cells in the presence of both the compounds, either in combination or alone, along with radiation. A combination of HNHA, sorafenib, and radiation was inhibited cell proliferation more effectively than either agent alone with radiation (Figure 3A,C), in a dose-dependent type (Figure 3B,D). Furthermore, the combination of HNHA, sorafenib, and radiation had a lower half maximal inhibitory concentration (IC_50_) than that of HNHA or sorafenib with radiation therapy in YUMC-A1 and YUMC-A2 cells (Table 1). These results show that this polypharmacy may offer a novel clinical approach for targeting stemness, metastatic cancer, including drug-resistant ATC, with low dosage of anticancer drugs.

### 2.4. Combination of HNHA and Sorafenib with Radiation Was More Effective in Inducing Apoptosis and Cell Cycle Arrest in YUMC-A1 and YUMC-A2 Than HNHA or Sorafenib with Radiation

Next, we investigated the mechanism of the synergistic anticancer effects of the combination of HNHA, sorafenib, and radiation on YUMC-A1 and YUMC-A2 cell lines. We estimated the levels of expression of the markers of cell cycle (cyclin D1 and p21), ER stress (GRP 78 and CHOP), and apoptotic signaling pathways (Bcl 2 and cleaved caspase 3) on YUMC-A1 and YUMC-A2 by flow cytometry, immunoblot (whole cell lysate or cellular fractionation), and immunofluorescence analyses. The combination of HNHA and sorafenib with radiation significantly induced the sub-G_0_G_1_ population, resulting in the induction of apoptosis and cell cycle arrest in YUMC-A1 and YUMC-A2 cells (Figure 4A,B) and leading to ER stress-mediated apoptosis, cell cycle arrest, and strong inhibition of YUMC-A1 and YUMC-A2 (Figure 4C,D). Immunoblot analysis of the protein expression levels in YUMC-A1 and YUMC-A2 indicated that the combination of HNHA, sorafenib, and radiation induced the most marked increase in the levels of p21, GRP 78, CHOP, and cleaved caspase 3 (which are associated with cell cycle arrest and are ER stress and proapoptosis markers). In contrast, the combination treatment resulted in decrease in the levels of cyclin D1 and Bcl 2, which are positive regulators of cell cycle and anti-apoptotic, compared with responses to HNHA or sorafenib administration with radiation.

### 2.5. Combination of HNHA, Sorafenib, and Radiation Induced Cytochrome c Release from Mitochondria and Translocation from the Cytoplasm to the Nucleus in ATC Cells

DNA damage is known to induce apoptosis by releasing cytochrome *c* from mitochondria [33]. To assess the mechanism of apoptosis induced by the combination of HNHA, sorafenib, and radiation, we evaluated the expression of cytochrome *c* and caspases with immunofluorescence and immunoblot analysis (Figure 5). Results from the immunofluorescence indicated that cytochrome *c* was located and accumulated in the nuclear, thus proposing that combination of HNHA, sorafenib, and radiation caused apoptosis via a cytochrome-c-dependent pathway in undifferentiated thyroid cancer cell lines, YUMC-A1 (Figure 5A,C) and YUMC-A2 (Figure 5B,D). Immunoblot analysis after subcellular fractionation confirmed that cytochrome *c* was translocated into the nucleus after combination treatment with HNHA, sorafenib and radiation (Figure 5C,D). In summary, these results suggest the combination of HNHA, sorafenib, and radiation induces apoptosis via caspase- and cytochrome-*c*-dependent pathways in undifferentiated thyroid cancer cells.

### 2.6. Combination of HNHA, Sorafenib, and Radiation Significantly Suppressed Tumor Growth in a Mouse Xenograft Model

To estimate the synergistic in vivo anticancer efficacy of the combination of HNHA, sorafenib, and radiation, we developed a mouse xenograft model with YUMC-A1 and YUMC-A2, patient-derived ATC cell lines (Figure 6). Results showed that HNHA or sorafenib treatment with radiation marginally suppressed YUMC-A1 and A2 cell xenograft tumors, whereas the combination of HNHA, sorafenib, and radiation significantly induced tumor shrinkage (Figure 6A,B). Moreover, mice in the HNHA, sorafenib, and radiation combination treatment group had significantly smaller tumor volumes than those of mice treated with HNHA or sorafenib administration with radiation (Figure 6C,D). There was no evidence of systemic toxicity and mortality in any group. Change in body weight was not significantly different among the two groups (Figure 6E,F). As anti-apoptotic activity is crucial for the appraisal of oncogenesis and Bcl-2 serves as a critical marker of anti-apoptotic activity, we confirmed this marker by immunohistochemistry analysis of YUMC-A1 and YUMC-A2 cell xenograft tumors. Results indicated that the mice in the HNHA, sorafenib, and radiation treatment group proved the maximum diminish in Bcl-2 expression among all the groups (Figure 6G,H). These results were supported that the combination of HNHA, sorafenib, and radiation showed potent anticancer effects in cancer stem cells (CSCs) and undifferentiated cancer cell xenograft model.

Consequently, these results propose a promising new curative approach to treat patients at a high risk of cancer-related mortality.

## 3. Discussion

The incidence of thyroid cancer is the highest among ordinary endocrine-related malignancies [34], and its worldwide occurrence is steadily increasing, including in Korea [35]. In 2011, the age-factual cancer occurrence ratio was 81.0 per 100,000, according to the data curated in the Korea National Cancer Incidence Database. It has been reported that the occurrence of thyroid cancer increase steadily in both men and women, and it is the most well-worn cancer among women in Korea since 2009 [36]. High-resolution ultrasonography has served to early detection of asymptomatic small thyroid nodules [37] and resulted in a reduction in the size ratio of identified thyroid cancers [38,39]. Many surgically removed cases of thyroid cancer were mainly owing to expansion in cancers determined at 1 cm or less. As a result, an increase in the assessment of thyroid cancer via neck ultrasonography and treatment in the early phase have moderated thyroid cancer-related mortality. However, ATC still remains one of the most common drug-resistant cancers [40]. Thyroid cancer types can be classified into those originated from follicular cells (well-differentiated follicular and papillary cancer), which frequently have a serendipitous diagnosis, or into ATC. The latter is a clinically aggressive form of thyroid cancer with poor diagnosis and includes poorly differentiated thyroid cancer [6,8,41]. ATC is considered a refractory cancer subtype, owing to its drug resistance and aggressive behavior [8,14]. Studies have reported that the median OS in patients with ATC is only several months [42,43].

Therefore, there is an urgent need to search for novel clinical approaches for the treatment of ATC. In this study, we proposed a promising novel treatment strategy for some intractable diseases in future. Synergistic anticancer effects of drugs that inhibit non-overlapping cancer pathways are a reasonable approach to suppress cancer cell proliferation. Furthermore, it is possible to reduce the doses of drugs when administered in combination with radiation and thereby increase the therapeutic efficacy and reduce the side effects. Epithelial–mesenchymal transition (EMT) is known to be connected to therapeutic resistance and recurrence as well as the acquisition of stem cell like features and is adequate to provide differentiated normal and cancer cells with stem cell features [44,45,46,47]. CSCs (including ATC) are poorly differentiated, have stem cell-like features [48], and have the ability to evolve. Moreover, these cells are connected with metastasis and therapeutic resistance [24]. Based on these findings related to CSCs, ATC was indicated to present a high expression of markers of the EMT and fibroblast growth factor receptor (FGFR) signaling pathway. In this study, based on the results of NGS (Next Generation Sequencing) analysis, the ATC cell lines showed EMT-mediated drug resistance through the FGFR signaling pathway and were a subtype of CSCs. Consequently, we concentrated on the effect of inhibition of the ATC by composition of anticancer drug and HDAC inhibitor, which would be promising therapeutic agent for the treatment of ATC cells in combination with tyrosine kinase inhibitors (TKIs), including sorafenib. Sorafenib is a commonly used multi-kinase inhibitor which inhibits the activity of Ser/Thr kinase Raf (which play a significant role in tumor cell proliferation or signaling) and angiogenesis-related receptor tyrosine kinases, VEGFR2 and PDGFR [28,49,50,51,52]. Sorafenib was recently approved for the treatment of refractory thyroid cancer [53]. It has been reported that the HDAC inhibitors increase p21 expression in various cell types via promoter hyper-acetylation [54]. Decline in p21 expression has been shown to be influenced by the lethality of HDAC inhibitors and DNA injury agents in varied cancer subtypes such as thyroid cancer, RCC, and leukemia [26,55,56]. It has been suggested that p21 is cleaved by caspase-3 for DNA damage-mediated apoptosis [57]. Furthermore, sorafenib-mediated transcriptionally inhibition could result in the down-regulation of p21 expression. Due to these reasons, it could be interesting to research whether the sorafenib mediated down-regulation of p21 expression can origin the synergistic interaction between HNHA and sorafenib.

In this research, we indicated that the combination treatment of HNHA, sorafenib, and radiation had a lower IC_50_ in ATC than that of either drug alone with radiation. The mechanism radical to these synergistic anticancer effects of either agent on ATC cell lines involved trigger of cell cycle arrest and apoptosis. However, the results from this study should be interpreted taking into account the limitation that the cells were obtained from a small number of patients. Nonetheless, this new combination of anticancer drugs and radiation for effective suppression of the refractory cancer could have potential clinical application and needs to be further evaluated.

## 4. Materials and Methods

### 4.1. Study Design

This research was a retrospective, single-center analysis of patients with proved ATC (diagnosed between February 2013 and November 2019). The electronic medical documents of the available patients were reviewed to extract data on clinical characteristics, including age, prior treatment, tumor characteristics and treatment outcomes. All process involving patients were carried out in agreement with the ethical standards of institutional regulations and all applicable local/national regulations, and with the 1964 Helsinki declaration and its later amendments or comparable ethical standards. In agreement with the Bioethics and Safety Act of Korea, formal consent was not requested for this type of retrospective, observational analysis.

### 4.2. Patients

Patients with proved ATC who accepted the combination therapy were eligible for inclusion in the analysis. Pathologic conformation of ATC was requested to confirm the prognosis of ATC, either through surgery or through open biopsy or core needle biopsy. Patients were followed up for at least 1 year or until death happened.

### 4.3. Tissue Specimens

Fresh tumors were collected from patients with histologically and biochemical proven ATC who were treated at the Severance Hospital, Yonsei University College of Medicine, Seoul, Korea. Fresh tumors were collected throughout surgical excision of ATC metastatic and primary sites. For the purpose of this study, we chose one patient with advanced metastatic ATC.

### 4.4. Ethical Considerations

The research protocol was approved by the Institutional Review Board of Severance Hospital, Yonsei University College of Medicine (IRB Protocol: 3-2019-0281). Cell samples were acquired from patients at the Severance Hospital, Yonsei University College of Medicine, Seoul, Korea.

### 4.5. Statistical Analysis

For the analysis of study results, categorical variables were described by proportion and frequency, while summary statistics (range, median) were used to describe continuous data. Survival curves were generated with the Kaplan–Meier method founded on the log-rank test. As this was a retrospective analysis, no regular statistical comparisons were carried out.

### 4.6. Tumor Cell Isolation and Primary Culture

After resection, tumors were kept in normal saline with antibiotic and antifungal agents and transfered to the laboratory. Further protocol details are recounted in our previous article [48].

### 4.7. Preparation of DNA

FFPE (Formalin-Fixed and Paraffin-Embedded) DNAs were isolated with the QIAamp DNA FFPE Tissue Kit (Qiagen, Valencia, CA, USA), pursuant to the manufacturers’ instructions. Initial QC tests of FFPE DNA were carried out with electrophoresis on 1% agarose gels and the Qubit dsDNA HS Assay Kit used the Qubit 2.0 fluorometer (Life Technologies, Carlsbad, CA, USA), pursuant to the manufacturers’ manual.

### 4.8. Preparation of Libraries

Libraries were prepared used the SureSelect XT protocol (Agilent Technologies, Santa Clara, CA, USA) used Custom Panel by the Macrogen (Macrogen, Seoul, Korea), and their quality was checked with the 2100 Bioanalyzer (Agilent). Further protocol details are recounted in our previous article [58].

### 4.9. Analysis of DNA Sequences

The adapter sequences were eliminated by fastp (Chen, 2018). Trimmed reads were aligned to the reference genome (GRCh37/hg19) with BWA-MEM (Li, 2013). Poorly mapped reads that had mapping quality (MAPQ) below 20 were eliminated with Samtools version 1.3.1 (Li et al., 2009). Duplicated reads were removed with Sambamba markdup (version 0.6.7) (Tarasov, 2015). Base quality of deduplicated reads was recalibrated with GATK BaseRecalibrator. Somatic mutations, including single nucleotide variants (SNVs) and small insertions and deletions (INDELs), were identified with MuTect2 algorithm (Cibulskis et al., 2013) [59,60,61]. Further protocol details are recounted in our previous article [62].

### 4.10. Cell Culture

Patient-derived ATC cells were acquired from the patient and grown in RPMI-1640 medium with 10% FBS (Authentication by short tandem repeat profiling/karyotyping/isoenzyme analysis). Mycoplasma contamination was invariably tested used the Lookout Mycoplasma PCR Detection Kit (MP0035, Sigma-Aldrich, MO, USA).

### 4.11. Cell Viability Assay

Cell proliferation was measured using the 3-(4,5-dimethylthiazol-2-yl)-2,5-diphenyl tetrazolium bromide (MTT) assay. Further protocol details are recounted in our previous article [62].

### 4.12. Irradiation

For in vitro tests, YUMC-A1 and YUMC-A2 cells were irradiated used Faxitron X-ray machine (Faxitron Bioptics, Tucson, AZ, USA) at 3–5 Gray (Gy), followed by treatment with HNHA or sorafenib alone and the combination of HNHA / sorafenib. For this experiment, mice were treated with the Small Animal Radiation Research Platform (SARRP) [High-resolution, small animal radiation study platform with x-ray tomographic guidance capabilities [62]. The tumors were irradiated with a circular beam of 1cm diameter with three successive daily fractions of 3 Gy.

### 4.13. Flow Cytometry Analysis of Cell Cycle

Cells were treated with HNHA and sorafenib alone or in combination with radiation in RPMI-1640 medium including 10% FBS for 40 h, harvested by trypsinization, and fixed in 70% ethanol. Further protocol details are recounted in our previous article [48].

### 4.14. Immunofluorescence Analysis and Confocal Imaging

The expression of cytochrome *c* was analyzed by immunofluorescence staining. Primary antibody was used anti-cytochrome *c* (1:25; Abcam, Cambridge, UK) in 5% bovine serum albumin in PBS. Further protocol details are recounted in our previous article [62]. Images were acquired and analyzed a confocal microscope (LSM Meta 700; Zeiss, Oberkochen, Germany) and Zeiss LSM Image Browser, version 4.2.0121.

### 4.15. Cellular Fractionation

Cellular fractions were arranged with the NEPER Nuclear and Cytoplasmic Extraction kit (Thermo Scientific, 78833, Waltham, MA, USA) in conformity with the manufacturer’s instructions. Further protocol details are recounted in our previous article [26].

### 4.16. Immunoblot Analysis

Primary antibodies against GRP 78 and CHOP bought from Cell Signaling Technology (Danvers, MA, USA), p21, Bcl 2, cytochrome c, and histone H2B (bought from Abcam), and cyclin D1, caspase 3, and β-actin (bought from Santa Cruz Biotechnology, Santa Cruz, CA, USA) overnight at 4 °C. Further protocol details are recounted in our previous article [62].

### 4.17. Human Thyroid Cancer Cell Xenograft

Patient-derived ATC cells (5.4 × 10^6^ cells/mouse) were cultured in vitro and then injected subcutaneously into the upper left flank region of female NOD/Shi-scid, IL-2Rγ KOJic (NOG) mice. Further protocol details are recounted in our previous article [62]. All experiments were confirmed by the Animal Experiment Committee of Yonsei University.

### 4.18. Immunohistochemistry

Primary monoclonal antibodies against Bcl 2 (Abcam), diluted with PBS (1:100), overnight at 4 °C. All tissue sections were counterstained with hematoxylin, dehydrated, and then mounted. Further protocol details are recounted in our previous article [62].

### 4.19. Statistical Analysis

Statistical analyses were carried out with GraphPad Prism software (GraphPad Software, Inc., La Jolla, CA, USA). Immunohistochemistry results were subjected to one-way analysis of variance, followed by a Bonferroni post hoc test. Values were indicated as means ± SEM. *p* values < 0.05 were considered as statistically significant.

### 4.20. Image Analysis

The MetaMorph 4.6 software (Universal Imaging Co., Downingtown, PA, USA) was used for computerized quantification of immunostained target proteins.

## 5. Conclusions

Synergistic anticancer activity of the HNHA, sorafenib, and radiation therapy was more effective than treatment with HNHA or sorafenib alone with radiation in patient-derived ATC. These findings can be useful to design future rational clinical studies in patient with ATC to develop effective therapies.

## Figures and Tables

**Figure 1 ijms-22-00536-f001:**
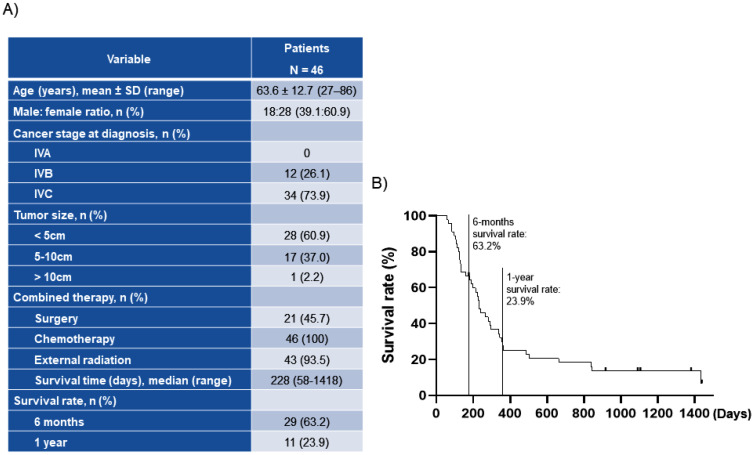
Information of ATC patients: (**A**) Patient characteristics and clinical features; (**B**) Survival rates of patients with anaplastic thyroid cancer. The cancer stage was determined according to the 8th edition of the AJCC/UICC by the American Joint Committee on Cancer and the International Union Against Cancer.

**Figure 2 ijms-22-00536-f002:**
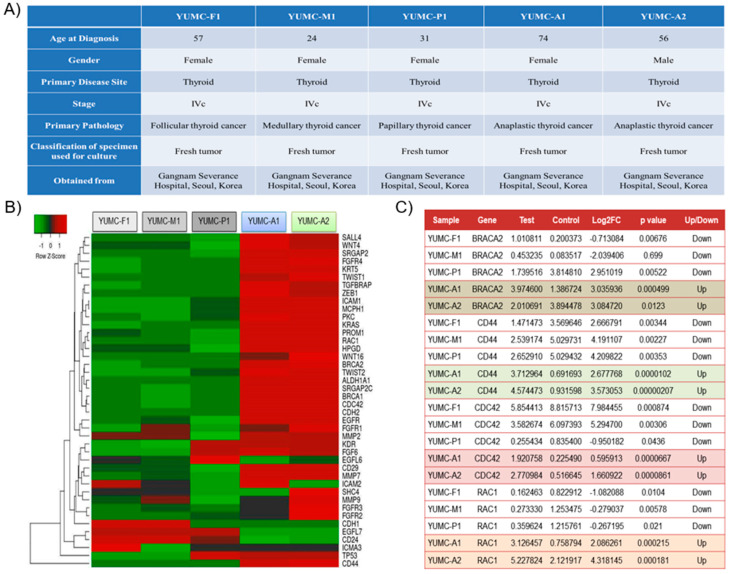
Characteristics of all thyroid cancer cell lines examined: (**A**) Characteristics of patient-derived subtypes of thyroid cancer cell lines, including viability after drug treatment of all thyroid cancer cell lines examined; (**B**) Hierarchical clustering of annotated genes revealed distinct gene expression. Gene expression profiles of various subtypes of patient-derived thyroid cancer cells; (**C**) The top four upregulated and downregulated differentially expressed genes (DEGs) for various subtypes of thyroid cancer cells.

**Figure 3 ijms-22-00536-f003:**
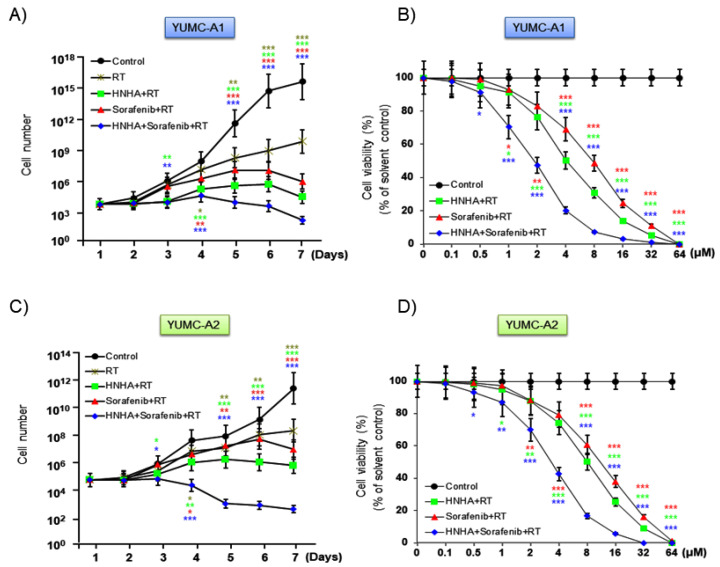
Synergistic anticancer effect of HNHA, sorafenib, and radiation on patient-derived thyroid cancer cells compared with the effects of each agent alone with radiation. Cell viability and proliferation assay of HNHA and sorafenib combined, each agent alone, or HNHA and sorafenib combined with radiation on patient-derived ATC cell lines. YUMC-A1 and YUMC-A2: (**A**,**B**) YUMC-A1 and (**C**,**D**) YUMC-A2,. Points indicate mean percentage of the value observed in the solvent-treated control. All experiments were repeated at least three times. Data represent means ± SD. * *p* < 0.05, ** *p* < 0.01, and *** *p* < 0.005 vs. control.

**Figure 4 ijms-22-00536-f004:**
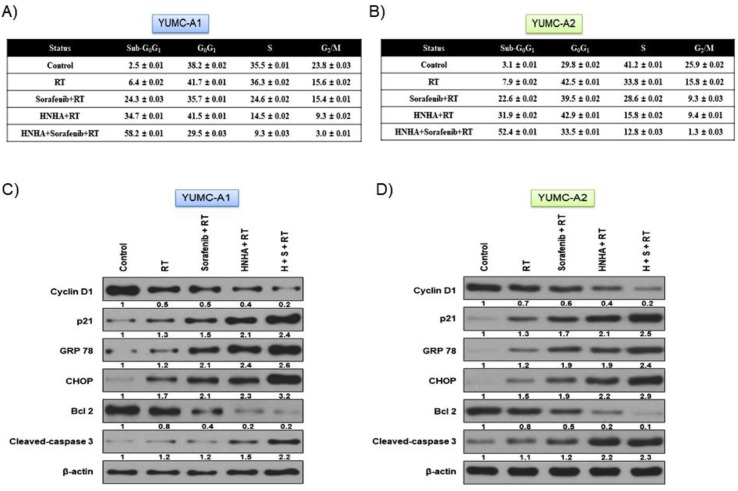
Cell cycle arrest and apoptosis analysis by quantitation of DNA content with propidium iodide and immunoblot analysis: (**A**,**B**) Cell cycle arrest and apoptosis were significantly induced by the combination of HNHA, sorafenib, and radiation in patient-derived ATC cell lines, YUMC-A1 and YUMC-A2. Cells were exposed to the indicated inhibitors, harvested, and stained with propidium iodide before analysis by flow cytometry and FlowJo v8; (**C**,**D**) Immunoblot analysis of the markers of cell cycle, apoptosis, anti-apoptosis, and ER stress on patient-derived ATC cell lines, YUMC-A1 and YUMC-A2.

**Figure 5 ijms-22-00536-f005:**
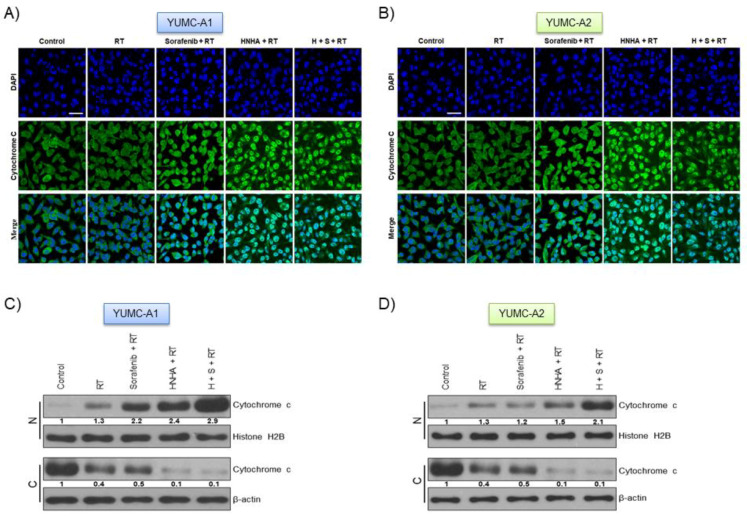
Synergistic anticancer effect of HNHA, sorafenib, and radiation was most induced nuclear-translocatyion of the cytochrome *c* on patient-derived ATC cell lines. Immunofluorescence (**A**,**B**) examined at ×400 magnification; scale bar, 20 μm and subcellular fractionation (**C**,**D**) analysis. Cytochrome *c* was most translocated and accumulated in the nucleus by HNHA, sorafenib, and radiation treatment group on patient-derived ATC cell lines: (**A**,**C**) YUMC-A1 and (**B**,**D**) YUMC-A2.

**Figure 6 ijms-22-00536-f006:**
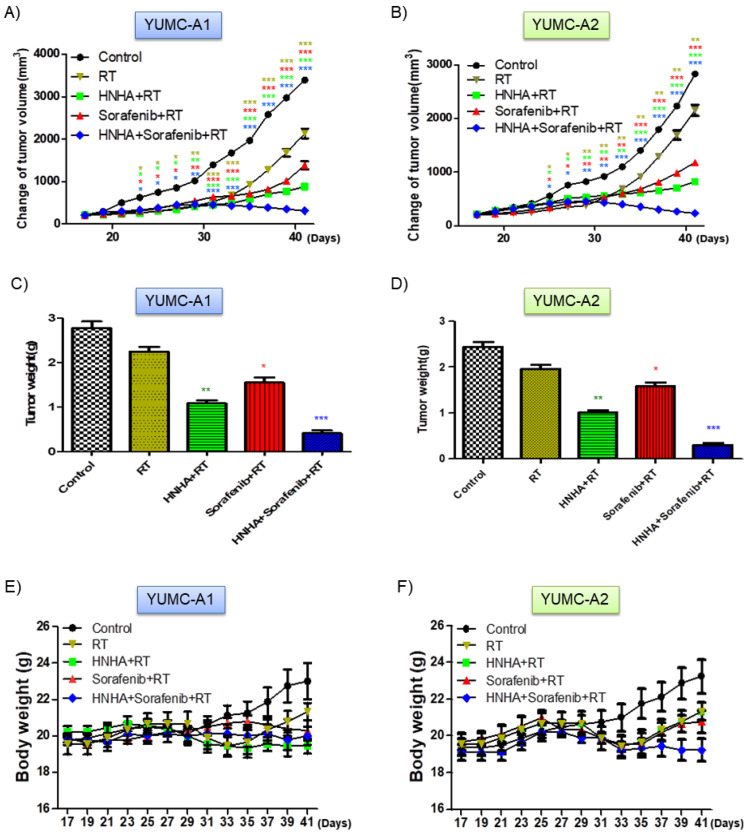
Synergistic anticancer effect of HNHA, sorafenib, and radiation induced the highest tumor shrinkage of ATC cancer cell xenografts in vivo: (**A**,**B**) Combination of HNHA, sorafenib, and radiation induced a significant inhibition of tumor progression than HNHA or sorafenib with radiation or radiation in patient-derived ATC, YUMC-A1 and YUMC-A2 cell xenografts (n = 10 mice/group). Data represent the mean tumor volumes. Co-treatment with HNHA and sorafenib with radiation showed the maximum reduction in tumor weight of the dissected tumor weight in patient-derived ATC, YUMC-A1 (**C**) and -A2 (**D**) cell xenografts. The compounds had no significant effect on body weight of patient-derived ATC, YUMC-A1 (**E**) and -A2 (**F**) cell xenografted mice. (**G****,H**) Immunohistochemical analysis of Bcl-2 proteins in tumor tissues following the indicated treatments. Examined at ×400 magnification; scale bar, 80 μm. Each assay was performed in triplicate and representative images are displayed. * *p* < 0.05, ** *p* < 0.01, and *** *p* < 0.005 vs. control. MetaMorph 4.6 image-analysis software was used to quantify the immunostained target protein.

**Table 1 ijms-22-00536-t001:** IC_50_ values for the combination of HNHA, sorafenib, and radiation therapy (RT) in YUMC-A1 and YUMC-A2 cells. Each data point signifies the mean of three independent MTT assays, performed in triplicate. SEM, standard error of the mean. MTT, 3-(4,5-dimethylthiazol-2-yl)-2,5-diphenyltetrazolium bromide. * is lowest half maximal inhibitory concentration.

Cell Line	Hisopathology	Animal	Cell Proliferation IC_50_ (μM)				
			HNHA	Sorafenib	HNHA + RT	Sorafenib + RT	HNHA + Sorafenib + RT
YUMC-A1	Thryoid, anaplastic	Human	16.19 (±0.6)	10.17 (±0.6)	6.71 (±0.1)	7.12 (±0.9)	2.74 (±0.6) *
YUMC-A2	Thyroid, anaplastic	Human	18.16 (±0.4)	12.14 (±0.5)	8.14 (±1.3)	10.33 (±0.7)	4.08 (±0.2) *

## Data Availability

Data is contained within the article.

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
