# Peer review of "Synergistic Anticancer Activity of N-Hydroxy-7-(2-Naphthylthio) Heptanomide, Sorafenib, and Radiation Therapy in Patient-Derived Anaplastic Thyroid Cancer Models"

_ijms, 2021, doi:10.3390/ijms22020536_

Round 1

Reviewer 1 Report

This study presents a promising innovative approach to treatment of ATC. The major limitation of this work is the small number of ATC patients. Some minor questions need to be addressed before considering the work for publication

Line 72 All patients received chemotherapy with paclitaxel or Adriamycin – what was the number of patients treated with paclitaxel or Adriamycin?

Figure 1. Cancer stages at diagnosis  – is this AJCC classification? Please include this information into the description of the figure

Line 156 - DNA injury

Line 323 - some references are missing in section 4.9. Analysis of DNA Sequences

Line 184 - Were there any other adverse effects of the treatments used?

Author Response

Reviewer 1.

Comments and Suggestions for Authors

This study presents a promising innovative approach to treatment of ATC. The major limitation of this work is the small number of ATC patients. Some minor questions need to be addressed before considering the work for publication

I don't know how to thank you enough for reviewing our manuscript. I agree with you completely. I have made the suggested correction. Thank you again for your review. I hope you are always healthy and happy!!

Line 72, All patients received chemotherapy with paclitaxel or Adriamycin – what was the number of patients treated with paclitaxel or Adriamycin?

Reply: Thank you for your comment. All patients received chemotherapy with paclitaxel, and 3 patients previously received adriamycin at another medical institution, and 93.5% of patients received radiation therapy. I have made the suggested correction, line 72.

Figure 1. Cancer stages at diagnosis – is this AJCC classification? Please include this information into the description of the figure.

Reply: Thank you for your comment. The cancer stage was determined according to the 8th edition of the AJCC/UICC by the American Joint Committee on Cancer and the International Union Against Cancer. I have made the suggested correction, legend of ‘Figure 1’.

Line 156 - DNA injury

Reply: Thank you for your comment. I have made this correction.

Line 323 - some references are missing in section 4.9. Analysis of DNA Sequences

Reply: Thank you for your comment. I have added some references. Thank you again for your comment.

Line 184 - Were there any other adverse effects of the treatments used?

Reply: Thank you for your comment. H&E staining was performed on heart and liver sections, but the toxicity problems were not indicated.

Reviewer 2 Report

This research paper clearly shows synergistic effect of N-hydroxy-7-(2-naphthylthio) heptanomide (HNHA), sorafenib (S), and radiation combination therapy in patient-derived anaplastic thyroid cancer (ACT) models. The data presented in this paper in support of their claim is very clear. However, there are some minor suggestions to enhance the quality of data presented in this paper.

  1. Qualitative data from western blot can be supplemented with quantitative analysis (Fig 4 and 5).
  2. Interestingly, the weight reduction of the tumor in radiation therapy group (RT) seems contradict the change in tumor volume data as compare to control. This has to be discussed. 
  3. Figure 3.  64uM dose of  both the drug alone with RT (HNHA+RT; S+RT) seems to be equally effective in promoting cell death in ACT cell lines as that of proposed combination therapy (HNHA+S+RT). This has to be discussed with respect to physiological dose given to patient. why this dose alone may or may not be useful when considering combination therapy proposed in this paper.
  4.  Figure 5, Immunofluroscence data presented in figure 5A and 5B does not corresponds to western blot data presented in this figure. Kindly use representative image to reflect Cytochrome C status in all treatment groups.

Author Response

Reviewer 2.

Comments and Suggestions for Authors

1. This research paper clearly shows synergistic effect of N-hydroxy-7-(2-naphthylthio) heptanomide (HNHA), sorafenib (S), and radiation combination therapy in patient-derived anaplastic thyroid cancer (ACT) models. The data presented in this paper in support of their claim is very clear. However, there are some minor suggestions to enhance the quality of data presented in this paper.

Reply: I don't know how to thank you enough for reviewing our manuscript. I agree with you completely. I have made the suggested correction. Thank you again for your review. I hope you are always healthy and happy!!

2. Qualitative data from western blot can be supplemented with quantitative analysis (Fig 4 and 5).

Reply: Thank you for your comment. I have changed figure of immunoblot analysis on figure 4 and 5, which were included quantitative analysis. Thank you again for your comment.

3. Interestingly, the weight reduction of the tumor in radiation therapy group (RT) seems contradict the change in tumor volume data as compare to control. This has to be discussed.

Reply: Thank you for your comment. I agree with you. But in my opinion, on the change of tumor volume data, radiation therapy group was look like induced tumor shrinkage for early phase (0~30days). However, tumor volume was significantly induced after 30 days until sacrificed. The measurement of dissected tumor weight may look like differ from change of tumor volume because measurement of dissected tumor weight is made after the sacrifice of mice. On YUMC-A1; control vs RT is 3200mm^3 / 2.9g vs 2100mm^3 / 2.3g, on YUMC-A2, control vs RT is 2900mm^3 / 2.5g vs 2200mm^3 / 2.2g.

4. Figure 3. 64uM dose of both the drug alone with RT (HNHA+RT; S+RT) seems to be equally effective in promoting cell death in ACT cell lines as that of proposed combination therapy (HNHA+S+RT). This has to be discussed with respect to physiological dose given to patient. why this dose alone may or may not be useful when considering combination therapy proposed in this paper.

Reply: Thank you for this comment. I agree with you completely. But in my opinion, the final goal of this research is looking for the best combination of anti-cancer drugs with the lowest concentration. Because, if it has the same effect, the lowest concentration of medications can lower the side effects (grade 3 or 4 of hand foot skin reaction, rash, hypertension, QT prolongation, Cardiac ischemia). Furthermore, it is well known that when these side effects occur, patient will stop or reduce medication [Ann Oncol. 2015 Oct; 26(10): 2017–2026]. In this study proposed that combination of HNHA, sorafenib and RT could be maximum anti-cancer effect with minimum dose.

5. Figure 5, Immunofluroscence data presented in figure 5A and 5B does not corresponds to western blot data presented in this figure. Kindly use representative image to reflect Cytochrome C status in all treatment groups.

Reply: Thank you for this comment. I agree with you completely. Follow expert your advice, figure 5A and 5B was corrected. Thank you again for your expert advice.